# Dairy Cows’ Temperament and Milking Performance during the Adaptation to an Automatic Milking System

**DOI:** 10.3390/ani13040562

**Published:** 2023-02-05

**Authors:** Jéssica Tatiana Morales-Piñeyrúa, Aline Cristina Sant’Anna, Georgget Banchero, Juan Pablo Damián

**Affiliations:** 1Programa Nacional de Producción de Leche, Estación Experimental INIA La Estanzuela, Instituto Nacional de Investigación Agropecuaria (INIA), Ruta 50 km 11, Colonia 70000, Uruguay; 2Departamento de Zoologia, Instituto de Ciências Biológicas, Universidade Federal de Juiz de Fora, Juiz de Fora 36036-900, MG, Brazil; 3Programa Nacional de Producción de Carne y Lana, Estación Experimental INIA La Estanzuela, Instituto Nacional de Investigación Agropecuaria (INIA), Ruta 50 km 11, Colonia 70000, Uruguay; 4Departamento de Biociencias Veterinarias, Facultad de Veterinaria, Universidad de la República, Ruta 8 km 18, Montevideo 13000, Uruguay

**Keywords:** animal welfare, parlour, personality, robot, stress

## Abstract

**Simple Summary:**

Dairy cows undergo an adaptative period during changes to a new milking system, such as automatic milking. During this period, cows could experience stress, affecting their health and productive performance. How cows cope with this period would depend on their individual characteristics, such as their temperament. In the present study, the relationships between temperament (measured by different handling yard tests), productive traits, and milking behaviours were evaluated. Cows classified as calm by the flight speed test exhibited more kicks and produced less milk during their first milkings with the automatic milking system than cows classified as reactive. Therefore, when the temperament was classified based on the flight speed test, calm animals presented greater difficulty in adapting to an automatic milking system than reactive ones. Understanding how cows adapt to new milking systems allows for the development of management strategies designed to improve animal welfare and the productive efficiency of the herds.

**Abstract:**

Adaptative responses of cows to an automatic milking system (AMS) could depend on their temperament, i.e., cows with certain temperament profiles could be able to cope more successfully with the AMS. The relationships between dairy cows’ temperament, behaviour, and productive parameters during the changeover from a conventional milking system (CMS) to an AMS were investigated. Thirty-three multiparous cows were classified as ‘calm’ or ‘reactive’ based on each of the temperament tests conducted: race time, flight speed (FS), and flight distance, at 5, 25, and 45 days in milk at CMS, then the cows were moved from the CMS to the AMS. During the first five milkings in AMS, the number of steps and kicks during each milking were recorded. The daily milk yield was automatically recorded. The number of steps did not vary by temperament classification, but the number of kicks per milking was greater for calm (0.45 ± 0.14) than for reactive cows (0.05 ± 0.03) when they were classified by FS (*p* < 0.01). During the first seven days in the AMS, reactive cows for the FS test produced more milk than calm cows (36.5 ± 1.8 vs. 33.2 ± 1.6 L/day; *p* = 0.05). In conclusion, behavioural and productive parameters were influenced by cows´ temperament during the milking system changeover since the calm cows kicked more and produced less than the reactive ones.

## 1. Introduction

Depending on how animals adapt to new environments, they may experience physiological, biochemical, and behavioural changes [1,2]. The milking system changeover implies a new situation, where cows should learn and adapt to new handling. The behavioural responses to an unknown environment or situation (e.g., a new milking system) depend on how positively or negatively the animals perceive the novelty. To our knowledge, behavioural responses of cows during the changeover from conventional (CMS) to automatic milking systems (AMS) have been insufficiently investigated. This situation of a change of milking systems was previously associated with a stressful experience for dairy cows, which negatively affected the milk yield and quality [3], milking behaviours, heart rate [3], and udder health [4]. Furthermore, Weiss et al. [3] demonstrated that inexperienced cows need careful adaptation to the AMS to minimise production losses.

Cows’ ability to adapt to novel milking conditions could be dependent on individual behavioural differences [5,6], which could be related to cows’ temperament. Sutherland and Huddart [7] reported that the response of heifers to the first week of lactation depends on their temperament and training to milking, i.e., trained heifers had higher reactivity to milking compared to untrained ones, but only when they were calm animals based on the flight speed test (FS). Specifically, in AMS, interindividual differences in cows’ adaptation have been reported [8,9] since cows’ reactions and speed of learning in the AMS are widely variable. Nervous cows that frequently display undesirable behaviours (e.g., stopping in front of the gate box) might have to be culled in this system [10,11]. Despite that temperament has been previously related to cows’ productivity and behaviours during the early postpartum period of cows experienced with an AMS [12], the adaptation responses of cows that entered into an AMS for the first time after being milked in a conventional system remain unknown. In addition, how temperament affects this adaptation process and the most desirable temperament traits for cows using an AMS have not yet been investigated.

In Uruguay, installation of the AMS has been increasing in recent years. The first AMS in Uruguay was installed in the Instituto Nacional de Investigación Agropecuaria (INIA), La Estanzuela, Colonia, based mainly on a mixed-feeding system (pasture and supplementation). The challenges arising from a transition from CMS to an AMS in mixed-feeding systems has been poorly studied, despite the AMS gaining more attention from producers. Investigating the factors related to the productive losses and stress responses due to the change of system is critical to the development of handling strategies to reduce these losses and improve animal welfare. Thus, we hypothesise that the temperament tests in the yard would predict the difficulties of cows to adapt to the AMS, related to behavioural responses and milking performance. Therefore, the objective of the present study was to assess the relationship between the reactivity to different tests of temperament of dairy cows and their behaviour and productivity during the changeover from a CMS to an AMS.

## 2. Materials and Methods

### 2.1. Animals and Husbandry Conditions

The study was conducted using thirty-three multiparous Holstein cows (3.0 ± 1.2 lactations) at the dairy research station of INIA La Estanzuela (Colonia, Uruguay, 34°35′ S, 57°70′ W), which comprises two milking systems, CMS and AMS, 300 m apart. The CMS is an 11-aside herringbone parlour where cows were milked twice a day (04:00 and 15:00 ) simultaneously by two stockpersons. The AMS has two monoboxes and two feed supply boxes from GEA (GEA Farm Technologies, Bönen, Germany). In both systems, cows wore a collar with a transponder for individual identification. During the prepartum period, all sampled cows were housed in a single group, being fed a total mixed ration (TMR) (Table 1) and provided with water ad libitum. The cows had access to a concrete feed pad (1 m in front/cow), a resting area, and shade (4.5 m^2^/cow in each area) in a compacted earthen surface area. In both systems, all cows had the same feeding regimen in the postpartum period (Table 1). The diet consisted of concentrate (around 7.6 kg of DM/cows/d) offered at the milking box (in CMS and AMS) and at the feed supply boxes (only in AMS), access to pasture after afternoon milking (around 12.0 kg of DM/cows/d), and TMR (around 3.0 kg of DM/cow/d) after morning milking. The feed boxes were always accessible during the trial and all animals could access them equally. The pasture was a mix of *Medicago sativa, Trifolium repens, Dactylis glomerata, Festuca arundinacea*, and *Bromus catharticus*. In the CMS, the TMR was delivered on wooden mobile feed pads (1 m in front/cow) in a feeding paddock of compacted earthen surface with a total area (feeding and rest area) of 5 ha, with a shade area of 4.5 m^2^/cow. In the AMS, the same TMR was offered in a feeding paddock of 1.2 ha of compacted earthen floor, with a concrete feed pad (1 m in front/cow), and the cows could voluntarily leave at any time. In both systems, water was available ad libitum, from automatic water troughs.

Cows were moved from the CMS to the AMS at 52.2 ± 3.9 days in milk (DIM) and trained on their first day in the AMS. Briefly, on the day of changeover, cows were milked at 06:00 at the CMS and then moved to the AMS and fed silage on a feed pad, while cows that were already being milked in the AMS herd had been milked nearby. Before noon, all untrained cows were induced to walk at their own pace through the smart gates and the milking robot three times. This procedure was repeated in the afternoon by the same handlers (who have been previously trained in good livestock management practices), but in the third passage across the robot, the gate of the milking station was closed, and the cows were milked.

### 2.2. Temperament Assessment

Cows’ temperament (evaluated while cows were at the CMS) was assessed in the handling yard using race time (RT), FS, and flight distance (FD) tests (details are provided below). The tests were performed on the same day between 15:00 and 17:00, in the order described above. All tests were conducted three times (mean ± SD) at the 7.3 ± 3.2, 25.4 ± 3.4, and 44.9 ± 11.0 DIM at the CMS. All behavioural tests were conducted by the same trained observer.

Race time was measured using a method modified from Pajor et al. [14]. One observer registered the time (in seconds) required to move each cow through the single-file race and enter the squeeze chute and the stimulus required to move the cow. Briefly, one cow at a time was moved from the holding pen to the entrance of the single-file race, which measured 5.95 m long, 0.80 m wide, and 1.26 m high, with walls made from solid wood. Cows were allowed 30 s to move through the race without any stimulus. If the cow did not move through the race, one handler applied increasing levels of stimulus until it entered the squeeze chute. The scale resulted in: (0) no need of stimulus application, animal moves through the race within 30 s on its own, (1) operator approach and speaks in a gentle voice, (2) hit rump with the open hand, and (3) push and force the animal to advance.

Flight speed was measured using a method modified from Gibbons et al. [15]. The time taken for each cow to exit the squeeze chute and cover a distance of 2.7 m was recorded using a purpose-built device that measured the exit speed. The unit comprised two light beams and reflectors and one readout unit. Once the cow passed through both light beams, the exit time was recorded in seconds and converted into speed (m/s). The handler stood approximately 1 m away from the side of the crush and behind the shoulder of the animal during the test.

Flight distance was measured using a method modified from Waiblinger et al. [16]. The distance (in m) that cows allowed a nonfamiliar person to approach before expressing the first withdrawal response was recorded. The test was performed with each cow individually kept in a corral pen of 131 m^2^. After the cow exited the squeeze chute, the observer remained stationary on the opposite side of the pen entrance 8 m away from the cow, waiting for the cow to stand still before starting the test. Then, the person approached the animal slowly (one step per s), with their hands down and arms held close to the body. When the cow expressed any withdrawal reaction (i.e., the animal moved both forelimbs), the observer stopped and measured the distance to the cow’s nearest front hoof using an odometer (MW40M, Stanley, Stanley Black & Decker, Inc., New Britain, Ct, EE.UU.).

### 2.3. Behavioural and Productive Responses

After changeover from CMS to AMS, during the first five milkings in AMS, the time each cow took to enter the milking box (in seconds, s) was registered. One handler applied increasing levels of stimulus until the cow entered the box, defining a score of willingness: (0) nothing, animal moves into milking box within 30 s on its own, (1) approach and speak in a gentle voice, (2) hit rump with open hand, and (3) push and force animal to advance.

Additionally, the following milking behaviours were registered during milking cluster attachment and the milking procedure: number of steps (counting the hind leg elevations lower than 15 cm off the ground) and number of kicks (counting the hind leg elevations higher than 15 cm off the ground with fast movements towards the milker). Cows’ behaviours were recorded by a single trained observer who remained still at a 3 to 5 m distance from the milking machine in a place that was not visible to the cows.

The occupation time per cow in the milking unit (box time, in seconds) was defined as the time from when the cow entered the milking box until she left the box, which was obtained by the automatic system of the milking machine of the AMS. Handling time (in seconds) was calculated as the difference between box time and milking time (duration of the milking), so it included the time from the cow entering the milking box until the milking starts, including the time for teat detection, washing, stimulation, and pre-milking, and the time after the milking stops until the entrance gate is open to allow the next cow to enter (same definition as in [17,18]). Additionally, the daily milk yield (L/day) and milk flow (L/min) were automatically registered by sensors of the milking machine (CMS and AMS), during seven days before the changeover and during the first seven days in the AMS.

### 2.4. Statistical Analyses

All statistical analyses were carried out with SAS Systems programs (SAS version 9.4, SAS Institute Inc. Cary, NC, USA). Univariate analyses were performed to identify outliers and verify the normality of residuals.

Cows were categorised using the terciles of distribution for each temperament test within days in the CMS (7.3 ± 3.2, 25.4 ± 3.4, and 44.9 ± 11.0 DIM). The consistency of the categories of each test was analysed by chi-square tests in contingency tables (temperament classes vs. days), and we did not obtain differences in the temperament categories across the days of assessment (*p* > 0.90). Therefore, the means of temperament traits of three days of assessment were obtained, and then cows were classified using the first and thirds terciles of distribution, where the extreme classes represented the least and most reactive animals as follows: for RT: calm (mean time ≥ 30.1 s; *n* = 11) or reactive (mean time ≤ 14.6 s; *n* = 11), for FS: calm (mean velocity ≤ 0.9 m/s; *n* = 12) or reactive (mean velocity ≥ 1.2 m/s; *n* = 8), and for FD: calm (mean distance ≤ 2.7 m; *n* = 12) or reactive (mean distance ≥ 3.5 m; *n* = 11). The classes were included as fixed effects in the models described below.

The relationships between temperament with behavioural responses and milking performances of cows during their adaptation to the AMS were assessed. To evaluate the effects of the temperament classes on time to enter the milking box, milking time, handling time, box time, and the numbers of steps and kicks, generalised linear mixed models for longitudinal data were fitted, via PROC GLIMMIX. The number of kicks was transformed to log-scale (log x + 2). Fixed effects of temperament traits in classes (one trait per model), time of assessment (milkings in AMS, from the first to the fifth milking), and their interactions were included. The random effect of animal was considered as a repeated measure within the time of assessment.

To investigate the possible effects of the system changeover and the temperament classes on milk yield and milk flow, the daily milk yields and flows of the first seven days in the AMS were expressed as relative values (%) of the mean CMS yields and flows of the seven days before the changeover. Linear mixed models for longitudinal data were fitted using PROC MIXED of SAS. Temperament classes, time of assessment (days in AMS, from the first to the seventh day), and their interaction were included as fixed effects. The animal was considered as a repeated measure within the time of assessment, and in addition the sire and DIM were considered as random effects. Previous milk yield and flow (seven days before system change) were included as covariates with a linear effect.

Additionally, partial correlations between milking behaviours (steps, kicks, time to enter the milking box, handling time, milking time, and box time) and milking performance (milk yield and milk flow) were analysed by a multivariate ANOVA (MANOVA) procedure. These correlations were calculated using all observations, and the partial correlations were adjusted for DIM and MY. Post hoc analysis was performed using Tukey´s test. The α values were considered significant when ≤0.05, and as a trend when ≤0.10.

## 3. Results

### 3.1. Temperament and Behavioural Responses to AMS

Regarding the reactivity to temperament test, only the number of kicks was influenced by reactivity classes. There was a significant interaction between FD and time of assessment (from the first to the fifth milking in the AMS) (*p* = 0.001). For calm cows, the number of kicks did not differ over time, while for reactive cows the number of kicks was greater at the first milking, then decreased from the second milking onwards (Figure 1). The difference between calm and reactive cows was observed at the first milking (Figure 1). Additionally, FS classes influenced the number of kicks (*p* < 0.01), and cows classified as calm had a greater number of kicks than reactive ones (Table 2). The RT was not related to the number of kicks (*p* > 0.10).

The time of assessment (from the first to the fifth milking in the AMS) only affected the number of steps (*p* < 0.05). The steps decreased from the first to the third milking, then increased until reaching the initial values at the fifth milking (Figure 2). Kicks and the time to enter the milking box were not different over time. None of the temperament tests were significantly related to the number of steps (Table 2). There was only a tendency for FS on the time to enter the milking box (*p* = 0.08), with calm cows displaying lower values than the reactive ones (Table 2).

Handling time was not different between temperament classes, time of assessment (from the first to the seventh day in AMS), or their interaction. The interaction between FS classes and time of assessment tended to affect the milking time (*p* = 0.09). The differences between calm and reactive cows were observed on day 5, when calm cows had a greater milking time than reactive cows. Additionally, FD classes tended to influence the milking time, with calm cows spending more time in milking (563.3 ± 75.2 s) than the reactive ones (400.2 ± 46.3 s) (*p* = 0.07). Box time was different according to FD classes, where calm cows had a greater time in the milking box (703.2 ± 74.7 s) than the reactive ones (516.6 ± 47.5 s) (*p* = 0.05).

### 3.2. Temperament Traits and Productive Responses

During the first seven days in the AMS, daily milk yield varied over time (*p* = 0.04) (Figure 3a). For FS, reactive cows produced more milk than calm cows (36.5 ± 1.8 L vs. 33.2 ± 1.6 L, *p* = 0.05). The RT and FD were not related to daily milk yield. When the relative milk yield in the AMS was analysed, cows produced less milk after the system changeover. The average milk yield during the first seven days in the AMS was 93.0% ± 3.7% of the CMS milk yield. The magnitude of the loss differed across the seven days in the AMS (*p* = 0.02), but the cows never reached the productivity they had in the CMS (Figure 3b). Relative individual yield ranged from 72.0% to 112.3% of the CMS yield. For FS, reactive cows had more relative milk yield than calm cows (99.6% ± 3.4% vs. 91.4% ± 3.0 %, respectively, *p* = 0.05). The RT, FD, and the interaction between temperament and the time of assessment (days in the AMS) did not affect the relative milk yield.

There was a significant interaction between RT and the time of assessment (*p* = 0.03) and an interaction between FS and the time of assessment (*p* = 0.03) on milk flow during the first seven days in AMS. During the first days in AMS, reactive cows (by RT and FS tests) had different flow evolution over time than calm cows, indicating more variation over time for reactive cows (Figure 4a,b). Differences between calm and reactive cows were observed for the RT test (Figure 4a) only on day 7 in the AMS. For the FS test, calm and reactive cows had different milk flows only on day 5 in the AMS (Figure 4b). The FD did not affect milk flow (*p* > 0.10).

The relative milk flow did not vary as a function of the time of assessment (days in the AMS) (*p* = 0.60), being on average 83.3% ± 3.4% of the flow obtained in the CMS before the changeover procedure. Relative individual milk flow ranged from 58.7% to 105.1% of the previous CMS flows. There was a tendency for the interaction between RT and time of assessment (*p* = 0.08) and a significant interaction between FS and time of assessment (*p* = 0.02) in milk flow during the first days in the AMS. For RT classes, calm cows never reached CMS milk flow values, but reactive cows did so on day 7 (Figure 5a). When cows were classified by FS, reactive cows had different relative milk flow evolution over time than calm cows, demonstrating more variation over time (Figure 5b), but none of the two temperament classes reached the pre-changeover milk flow level.

### 3.3. Relationships between the Behavioural and Productive Measures of Adaptation

Low and negative partial correlations were found between the number of steps and milk flow (r = − 0.18; *p* = 0.04), and there was a negative tendency between the number of kicks and milk flow (r = − 0.16; *p* = 0.06), i.e., higher numbers of steps and kicks were weakly related to reduced milk flow in the AMS. Additionally, milk flow was positively correlated with milk yield (r = 0.38; *p* < 0.0001).

## 4. Discussion

To our knowledge, this is the first study evaluating the relationship between temperament tests, behaviour, and productive responses of Holstein cows when relocated from a CMS to an AMS, both mixed-feeding systems. Cows need a few days to adapt to an AMS [3,6], but the adaptation to a new system could be more difficult for some cows than for others [5,19]. These individual differences could be determined by the cows’ temperament. In the present study, it was evidenced that temperament (calm vs. reactive) influenced some adaptative responses, such as kicking in the milking box, milk yield, and flow.

When cows are moved from a CMS to an AMS, they exhibit high levels of stress-related behaviours (stepping, kicking, urination, defecation), which decrease over time as the cows habituate to the AMS [6]. It is reported that in less than 24 h after the system changeover, stepping and kicking decrease, and then remain constant [6]. In the present study, the number of steps and kicks did not have a clear change over the successive milkings in the AMS, possibly due to the short duration of the evaluation (five milking days) and/or the low frequencies of these behaviours. The temperament classes influenced the milking reactivity in the AMS, since the number of kicks varied according to some temperament tests. For FD, reactive cows kicked more than calm cows, but only on the first milking day, then decreased their reactivity. The higher reactivity at the first milking in the AMS may be expected because the milking system represented a novel routine, and therefore, reactive cows would react more to the novelty. However, their reactivity decreased over the next days, maybe because the AMS implies little contact with humans and FD is a test that evaluates the fear of humans. The system conditions could have favoured the decrease in reactivity more in reactive cows than in calm ones. Regardless of the time of assessment (i.e., milkings in AMS), cows classified as calm by FS presented more kicks than reactive ones. A similar result was reported by Sutherland and Dowling [20], working in a CMS, who suggested that heifers with a lower exit speed kicked more during milking than the faster animals. However, another study in a CMS with multiparous cows reported that FS was not related to kicks [21]. Therefore, there was no clear and direct pattern of association between handling temperament in the yard with reactivity during milking. Less reactive animals during handling tests (e.g., FS) would be more reactive during milking due to different handling situations, expressing different temperament dimensions.

In our study, the time to enter the milking box did not decrease over the evaluated milkings. Weiss et al. [3] reported in cows under similar conditions of traffic, but with cows in confinement, that after the third day in the AMS, cows were able to enter the milking box without physical forces. In the present study, the five milkings (i.e., three days) could have been not enough to observe differences in this behaviour. However, we observed that reactive cows classified by FS took more time to enter the milking box than calm cows. The FS has been associated with a fear of restraint situations but could also be related to the motivation of the cows to enter the milking box in an AMS. Perhaps the cows had similar perceptions about both places. Conversely, the FD classification was not associated with time to enter the milking box, which could be related to the fact that the test probably indicates a fear of humans and not of the milking box. Considering that the AMS usually implies scarce human contact, the FD would not be a good indicator of temperament, or it does not have the same interpretation of cows in this system as cows in a CMS. Although Brouček and Tongel [11] suggested that avoidance of the milking box in the AMS can be related to unfavourable temperament, there are no studies linking the reactivity to the test of temperament in yards of cows with their behaviour of entrance into the milking box in AMS.

The milk yield and flow were affected by system changeover, being lower in AMS than in CMS, which had already been reported in previous studies [3,6]. During the first seven days in AMS, the cows failed to reach the CMS milk production and flow level. Unlike primiparous cows, who have better adaptation success in an AMS than in a CMS, as evidenced by higher milking frequency [22] and milk production [23], the multiparous cows have negative effects on milk production during the adaptation period in an AMS [8]. In the study of Jacobs and Siegford [6], cows had milk values similar to the CMS at four days post-change. In our results, similarly to Weiss et al. [8], cows did not return to CMS milk values before the first 10 days in the AMS. The milk production level could be an explanation for the differences among the study of Jacobs and Siegford [6] and ours since their cows had a lower milk yield level than the cows in our study. It is possible to infer that their cows could reach CMS milk values faster than ours.

In our study, we had a wide range of relative individual yields, which could be related to cows’ temperament. In fact, cows classified as calm by FS lost more milk and produced less than reactive ones. Individual differences in milk and flow losses have also been reported by Weiss et al. [3]. Different than our results, the milk performance in a novel environment (in their case a CMS) was worse for reactive than for calm cows [21]. The temperament tests may not have the same relationship with productivity in an AMS as in a CMS, as suggested by Morales-Piñeyrúa et al. [12]. In this sense, this study highlights the need to evaluate the reactivity of the animals using other indicators that do not imply contact with humans in the adaptation to novel management systems, such as the AMS.

Conversely, calm cows kicked more than reactive animals; therefore, greater reactivity to milking could have resulted in incomplete milking and consequently less milk yield [24], as well as longer milking times (calm cows had the greatest time in the milking box compared to the reactive ones). This would agree with the results that kicking cows tended to exhibit a lower milk flow (correlations results), and less milk flow is reflected in less milk yield. Kicking is an indicator of discomfort caused by low milk flow [11]. On the other hand, reactive cows had peaks of milk flow higher than calm cows, which could have provoked, on average, a greater milk yield. The milk flow has been reported as a promising trait for describing milking ability in the AMS [18], such as a measure of adaptation to milking [25]. However, in the present study, the milk flow decreased and continued to be low in the AMS. It is possible that more days would be necessary to restore the milk flow after system changeover.

In our study conditions, of the three temperament tests used, we suggest that flight speed is the most practical to perform on-farm, being able to predict behaviour and productive patterns at milking during the adaptation period at an AMS in mixed-feeding systems. Calmer cows, as seen by FS, were more sensitive to system change than reactive cows (more milking reactivity and worst productivity). Therefore, FS tests could be used on commercial farms to characterise individual animals, helping to select cows that better fit to the system, and/or to tailor cow management according to their individual needs. However, we caution that this test may be more linked to the human–animal relationship, and in the AMS this factor could be non-important. Future research should determine temperament related to the fear of novel objects or situations (such as arena tests or novel object tests), and how these could affect the adaptive responses to the AMS. We also caution that our milking behaviour results may not be strictly extrapolated to all situations because of the low sample size used in the experimental farm conditions. All these topics require further analyses, including a wider range of adaptability parameters, such as milk quality indicators.

## 5. Conclusions

Cows’ temperament influenced their milking reactivity and productivity during the changeover from a CMS to an AMS. Calmer cows kicked more and produced less milk than reactive ones. Therefore, calm cows could have more difficulties adapting to the AMS. Additionally, in our study conditions, relocation of cows from the CMS to the AMS did not generate a prolonged behavioural reaction, since responses of fear or stress (steps, kicks, the time to enter the milking box, box time, handling time) did not increase over time. However, the milk flow and production did not reach the CMS values in seven days post-system change.

## Figures and Tables

**Figure 1 animals-13-00562-f001:**
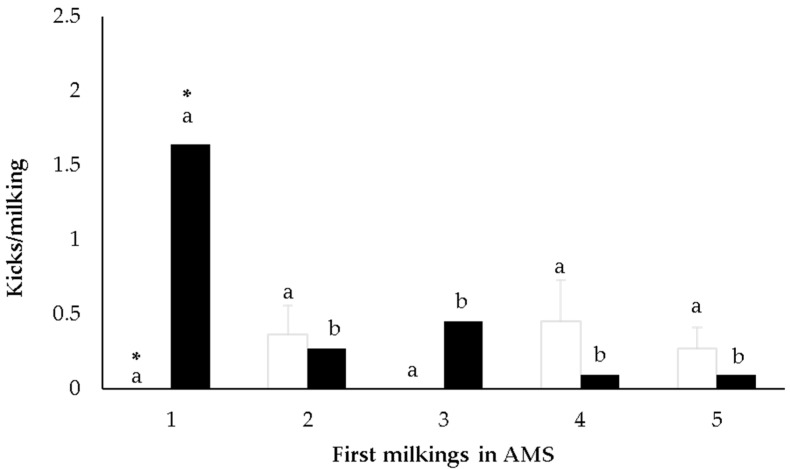
Number of kicks of multiparous Holstein cows during the first five milkings in the automatic milking system (AMS) according to flight distance test classes: calm cows (white) or reactive cows (black). Differences between milkings within each temperament class are indicated with different letters, and differences between temperament classes within the same milkings are indicated with * (*p* ≤ 0.05).

**Figure 2 animals-13-00562-f002:**
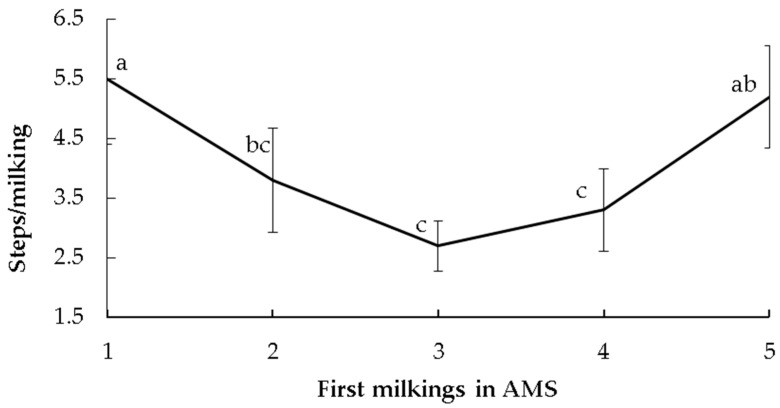
Number of steps of multiparous Holstein cows during the first five milkings after changeover from a conventional milking system to an automatic milking system (AMS). Differences between milkings are indicated with different letters (*p* ≤ 0.05).

**Figure 3 animals-13-00562-f003:**
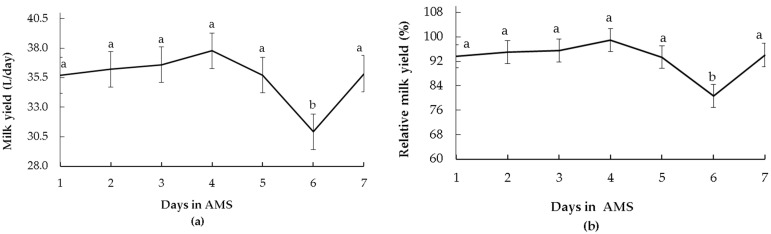
(**a**) Total milk yield (mean ± SE) and (**b**) milk yield (mean ± SE) expressed as relative values of conventional milking system (CMS) results (100% = mean CMS yields of seven days prior to the system changeover) of multiparous Holstein cows during seven days after the changeover from the CMS to the automatic milking system (AMS). Differences between days are indicated with different letters (*p* ≤ 0.05).

**Figure 4 animals-13-00562-f004:**
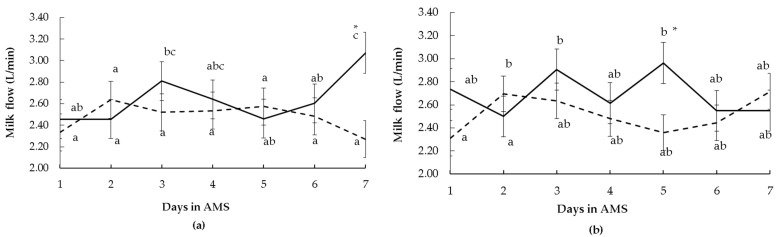
Milk flow (mean ± SE) of multiparous Holstein cows during seven days after the changeover from the conventional milking system (CMS) to the automatic milking system (AMS) according to (**a**) race time and (**b**) flight speed test in the CMS (calm: curt line, reactive: solid line). Differences between days are indicated with different letters, and differences between temperament classes within the same days are indicated with * (*p* ≤ 0.05).

**Figure 5 animals-13-00562-f005:**
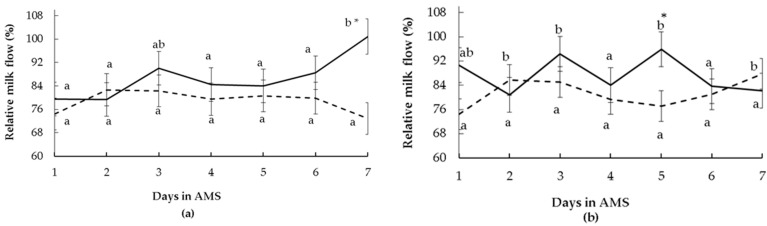
Milk flow (mean ± SE) expressed as relative values of the conventional milking system (CMS) results (100% = mean CMS yields of seven days prior to the system changeover) of multiparous Holstein cows during seven days after changeover from the CMS to the automatic milking system (AMS) according to (**a**) race time and (**b**) flight speed test at CMS (calm: curt line, reactive: solid line). Differences between days are indicated with different letters, and differences between temper-ament classes within the same days are indicated with * (*p* ≤ 0.05).

**Table 1 animals-13-00562-t001:** Ingredients (mean of kg DM/cow/d), chemical composition (% of total kg DM), and metabolizable energy (ME) of the total mixed ration (TMR), pasture, and concentrate offered during prepartum and postpartum periods to Holstein cows in an automatic milking system.

Ingredients	Prepartum	Postpartum
TMR		
Corn silage	6.20	0.98
Barley straw	2.20	-
Pasture haylage	-	1.08
Corn grain	-	0.64
High-moisture corn grain	0.88	-
Soybean meal	1.40	0.58
Soybean hulls	0.60	0.41
Urea	0.08	0.02
Calcium carbonate	0.04	-
Sodium bicarbonate	-	0.02
Dicalcium phosphate Magnesium oxide	-	0.005
Prepartum vitamin–mineral mix ^1^	0.30	-
Postpartum vitamin–mineral mix ^2^	-	0.01
Pasture	-	12.0
Commercial concentrate	-	7.6
Total DM	11.7	23.3
Chemical composition		
Crude protein, %	14.6	20.2
NDF, %	37.4	36.9
ADF, %	23.6	21.1
ME ^3^ provided from TMR (MJ/kg DM)	29.3	10.9
ME provided from pasture (MJ/kg DM)	-	9.9
ME provided from commercial concentrate (MJ/kg DM)	-	11.9

^1^ Provided (per kg of DM of mix): 230 g of Ca, 20 g of P, 100 g of Mg, 130 g of Cl, 30 g of S, 0.8 g of Cu, 2.4 g of Zn, 12 mg of Se, 1.4 g of Mn, 5 mg of Co, 89,000 IU of vitamin A, 17,000 IU of vitamin D, and 3500 IU of vitamin E. ^2^ Provided (per kg of DM of mix): 140 g of Ca, 14 g of P, 30 g of Mg, 150 g of Na, 110 g of Cl, 597 ppm of inorganic Cu, 277 ppm of organic Cu, 1815 ppm of inorganic Zn, 853 ppm of organic Zn, 6 ppm of inorganic Se, 3 ppm of organic Se, 1090 ppm of Mn, 65,707 IU of vitamin A, 13,141 IU of vitamin D, and 298 IU of vitamin E. ^3^ Metabolizable energy was calculated by chemical composition of foods [13].

**Table 2 animals-13-00562-t002:** Mean ± SEM for different milking behaviours by temperament (classified by race time (RT), flight speed (FS), and flight distance (FD)) of multiparous Holstein cows during the first five milkings in an automatic milking system, after changeover from a conventional milking system.

Temperament Traits		Time to Enter Milking Box (s)	Steps (Number)	Kicks (Number)
RT	Calm	49.3 ± 4.9	3.9 ± 0.9	0.42 ± 0.16
	Reactive	42.2 ± 4.3	4.6 ± 0.9	0.18 ± 0.07
FS	Calm	52.9 ± 4.5 b	3.3 ± 0.6	0.45 ± 0.14 A
	Reactive	68.1 ± 7.2 a	2.9 ± 0.8	0.05 ± 0.03 B
FD	Calm	53.8 ± 5.5	2.3 ± 0.7	0.22 ± 0.08
	Reactive	55.3 ± 5.6	4.3 ± 0.6	0.51 ± 0.17

Difference between temperament classes within each temperament trait is indicated with different capital letters (*p* ≤ 0.05), and the tendency is indicated with lowercase letters (*p* ≤ 0.10).

## Data Availability

The data presented in this study are available upon request from the corresponding author. The data are not publicly available due to the privacy statute of INIA.

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
