# Peer review of "Dairy Cows’ Temperament and Milking Performance during the Adaptation to an Automatic Milking System"

_animals, 2023, doi:10.3390/ani13040562_

Round 1

Reviewer 1 Report

The study describes the impact of cows' temperament on adaptation to AMS in terms of behavior and production responses. Overall, the study is well-designed and fulfills the criteria to make space in the journal. Nonetheless, the authors should address some specific details, listed below.

L 66-77: in my opinion, the authors fail to mention in the structuring of the state of the art what is reported by the paper identified as doi.org/10.1016/j.rvsc.2022.04.001. Therefore, I invite the authors to review the introduction, placing emphasis on the differences between the cited research and the present one and on how the latter fits, innovatively, with respect to the focused topic knowledge. Thanks.

L 72-74: As I read, as well as what is addressed in the text, the hypothesis tested by the authors is only one in my opinion. Therefore, I invite the authors to review the use of the plural or, vice versa, to rephrase the sentence. Thanks.

L 91 (and through the text): as written, AM is readable as an acronym. Please report it as a.m. Thanks.

L 80-93: I would invite the authors to report in table form the formulation (quantity of each dietary ingredient) of the diet used during the observation period and the relative nutritional value. Similarly, the authors could detail the floristic composition of the pasture and the pasture management was done. I think this information is useful, even though the work focuses on non-nutritional aspects. Thanks.

L 84-85: I invite the authors to detail if the feeding boxes were always accessible during the trial and if all the animals could access them equally. Thanks.

L 97-100: although in brief, the authors could provide more details on how the untrained cows were handled to make them forced to move. The activities relating to the handling of the animals and the stock people's attitude could affect the behavior and restlessness of the animals, with consequent potential conditioning of the behaviors observed during the cows' behavior recording phase. Thanks.

L 166 (and through the text): according to the journal template, the p-value should be reported in italics and lowercase. Thanks.

L 356-357: given the relevance and tangentiality between dairy species of milk flow as an adaptive measure of milking in AMS, I invite the authors to update the references supporting the statement on lines 356-357, adding to the [22] also the following reference doi.org/10.3168/jds.2017-14157, addressing the same issue of the perspective of large ruminants other than cows. Thanks.

Author Response

The authors thank the reviewer for their help in improving the quality of the manuscript. We answered all reviewer’s questions and comments. The changes in the manuscript were marked up using the “change control”, and we present detailed responses to the reviewer`s comments in the word file.

Reviewer 2 Report

Comments to the author are found below. The specific sections of the manuscript referred to are highlighted in the attached.

SIMPLE SUMMARY

Line 16-26: This section requires significant grammatical correction.

INTRODUCTION

Line 47-48: This implies that if cows are moved to a worse environment they will adapt slowly, is this correct? This also implies that AMS are “better” than CMS- this is not necessarily true. I suggest rewording or deleting this sentence.

Line 59: please define FS

MATERIALS & METHODS

Line 80: How was your sample size determined? Please provide evidence of adequate power to detect meaningful differences

Line 141: Should the 0 score be “moves into AMS within 30 s on its own” instead of “moves through race”?

Line 179-180, Line 185 -187: The term “time of assessment” is not clear. Do you mean day of study/day on AMS?

RESULTS

Line 200: Time for what?

Figure 1, Figure 2: Y axis title should include units.

Table 1: Time to enter into milking box column should include units.

Line 213, Line 228-229, Line 252, Line 264, Line 267, Line 268: Please be more clear about what is meant by “time of assessment”.

Line 256- 257: Please reword to be more clear about what day 7 and day 5 of assessment is.

Line 269-270: Graph 5a seems to indicate the opposite, i.e. RT calm cows reached CMS milk flow levels at day 7.

Line 283: Please define the adaptation period. Are you referring to the 7 d after switching to AMS?

DISCUSSION

Line 307: Please be more clear about what is meant by “time of assessment”.

Line: 319- 320: Please revise this sentence to improve grammar.

Line 324: Please change DF to FD.

Author Response

(The authors gave the same response as above.)

Reviewer 3 Report

Regardless of the formulation of the general purpose of the research/study, it would also be worth writing what was the cognitive (scientific) goal of the research undertaken, and what was the utilitarian (useful) goal. In my opinion, the review of the state of knowledge in the Introduction should be summarized in the final part by formulating the research problem (The research problem is …). The research problem can be related to the indication of the gap in the current state of knowledge, which was one of the main reasons for undertaking a research study with cows in the field of adaptation to milking with the use of AMS.

In the Abstract, in lines 35-36, the Authors wrote "... the kicks were greater for calm (0.45 ± 0.14) than for reactive cows (0.05 ± 0.03)...". I don't understand what the numbers in parentheses mean if the units are not given. Is it the number of kicks per milking or per 24 hours? Or is it not about the number of kicks? I couldn't figure out what data was being compared here. Similar unit issues apply to the data on line 37 in the Abstract. Is it the amount of milk milked per cow per day or are other units included? In my opinion, these details need to be completed.

The authors indicated that the behavior of the cows was observed by a trained person. Didn't the authors consider the possibility of using cameras to record animal behavior instead of direct observation of cow behavior at the AMS stall? I think that this solution option can be used in further, future research. The use of cameras would be less strenuous compared to direct observations, and the recorded material can be used for repeated playback and observation of additional details in the experiment.

I would like to know if one limit value of selected behavior indicators was considered between the calm cow and reactive cow categories? The authors have given ranges for RT, FS and FD indicators, but these ranges do not have the same cutoff value.

In my opinion, the group of cows included in the study could have been described more precisely. What lactation were the cows in? More details could have been provided regarding body weight, size, udder shape and problems with udder shape that would make it difficult to attach teat cups to teats.

I suggest that in the Materials and Methods chapter, you write in what conditions the cows were kept in the barn. Was it a loose housing or free-stall barn? On what floor material did the cows lie in the lying area. Such and other information allows the reader to know whether the animal welfare conditions in the barn were met, not only during the experiment, but also during the production period.

In Figure 3a, the authors provided the Milk Yield unit in liters (L) on the ordinate axis (y). In my opinion, it would be worth specifying this unit more precisely. Was it in "L/day" or "L/milking"? To avoid any doubt, I suggest giving the unit in a more precise way. I have a similar concern with the ordinate (y) axes in Figures 1 and 2. In my opinion, it would be a good idea to write kicks/milking and steps/milking respectively.

How and with what devices was the milk flow of individual cows tested in the experiment? Milk flow and its changes in the case of AMS could certainly be analyzed using the AMS installation with specialized sensors. Whereas how milk flow was studied in CMS. Were special flow meters used for CMS milk flow studies? It is worth writing about it in the Materials and Methods chapter.

Was the kicking of the cows observed in the experiment a threat to the dropping of the teat cups during milking? Has this risk factor been considered in the study? Please reply.

Did the movement of the cow and the shifting of the cow from foot to foot in the robotic milking stall (AMS) contribute to the problems with connecting the teat cups to the udder? Were there any cases during the research that due to the nervous behavior of the cows, the robot was unable to connect the teat cups and after unsuccessful attempts the cow had to leave the milking stall? In this case, the cow must enter the milking stall again. It is worth writing about such details from the observation in the article to show the problems associated with the transition of cows from the CMS system to the AMS system.

I think that an important parameter for comparison, apart from milk yield of cows, is also milk quality expressed by TBC and SCC. Were these indicators also assessed? In general, the number of bacteria in milk (TBC) increases with the transition to AMS and this element is worth considering in the study and discussion. If studies have not been done in this area, there is an incentive to develop TBC and SCC analyzes in future experiments.

How can the results of research and observations be translated into recommendations for farmers who would like to equip their farms with AMS? I think it is worth writing about such practical aspects of research and observation results that increase the value of the experiment.

Author Response

(The authors gave the same response as above.)

Round 2

Reviewer 1 Report

Dear authors,
I have reviewed the revised version of your manuscript identified as animals-2161352. Taking into account the changes made, also in light of the suggestions of other reviewers, I believe that the manuscript has been significantly improved. Therefore, I have no doubt in suggesting the publication of the manuscript which, in my opinion, can take place in its current form.
Congratulations